# Deciphering the Biological Effects of Radiotherapy in Cancer Cells

**DOI:** 10.3390/biom12091167

**Published:** 2022-08-23

**Authors:** Zhou Lu, Xueting Zheng, Chenghe Ding, Zhiyan Zou, Yuanyuan Liang, Yan Zhou, Xiaoan Li

**Affiliations:** NHC Key Laboratory of Nuclear Technology Medical Transformation, Mianyang Central Hospital, School of Medicine, University of Electronic Science and Technology of China, Mianyang 621000, China

**Keywords:** cancer, radiotherapy, cell death, radioresistance, immunomodulating effects

## Abstract

Radiotherapy remains an effective conventional method of treatment for patients with cancer. However, the clinical efficacy of radiotherapy is compromised by the development of radioresistance of the tumor cells during the treatment. Consequently, there is need for a comprehensive understanding of the regulatory mechanisms of tumor cells in response to radiation to improve radiotherapy efficacy. The current study aims to highlight new developments that illustrate various forms of cancer cell death after exposure to radiation. A summary of the cellular pathways and important target proteins that are responsible for tumor radioresistance and metastasis is also provided. Further, the study outlines several mechanistic descriptions of the interaction between ionizing radiation and the host immune system. Therefore, the current review provides a reference for future research studies on the biological effects of new radiotherapy technologies, such as ultra-high-dose-rate (FLASH) radiotherapy, proton therapy, and heavy-ion therapy.

## 1. Introduction

Globally, malignant tumors have become major conditions that significantly compromise human health and lives. According to the GLOBOCAN estimates from the International Agency for Research on Cancer, a total of 19.3 million new cases of cancer and 10 million deaths were expected to occur in 2020 [1]. Radiation therapy (RT) is one of the most important treatment strategies against malignant tumor cells. Approximately 50% of patients with cancer are estimated to receive RT as part of their treatment, and it has been reported that 40% of patients are cured by the therapy [2,3]. However, the success of radiotherapy is threatened by the emergence of radioresistance by cancer cells and the RT-induced damage to normal cells.

Currently, there is an increasing understanding of the radiobiological effects and mechanisms of tumor cells in response to RT. This is an important projection toward the development of new therapeutic strategies and the realization of optimal treatment outcomes in patients with cancer [4]. Although some progress has been made in the tumor radiobiological response, there is still a need to elucidate more of the molecular mechanisms and biological signatures of tumor cells after exposure to radiation. The present review aims to introduce the major forms of cancer cell death programs that are induced by ionizing radiation. The current study summarizes the multiple molecular mechanisms involved in the cellular responses to radiation that lead to cell death or survival. In addition, various immunomodulatory effects of irradiated tumor cells are highlighted in the current study.

## 2. RT-Induced Cancer Cell Death

The main goal of RT is to destroy or slow tumor growth by using high-energy radiation, such as X-rays, gamma (γ) rays, electrons, protons, neutrons, and carbon ions. The efficacy of killing is influenced by a number of factors, including the type of radiation, the total dose, the fractionation rate, and the targeted organs. It is noted that different tumors show different levels of sensitivity to RT. Previous studies have shown that the primary intracellular target of RT is DNA [5]. RT triggers DNA damage through the direct deposition of ionizing energy into the DNA or via the production of free radicals. Consequently, the damaged DNA is efficiently detected and repaired through homologous recombination (HR) or non-homologous end-joining (NHEJ) mechanisms. At present, NHEJ is the preferred pathway for repairing RT-induced DNA damage. A high radiation-associated deletion burden was associated with poor survival and may be able to predict the sensitivity of recurrent cancer after RT [6]. Conversely, improper or inefficient mechanisms of DNA repair induce different manners of cell death, including mitotic catastrophe, apoptosis, and senescence (Figure 1). Comprehensive reviews on the molecular mechanisms of these cell death phenotypes have previously been studied and published [7,8]. 

The modes of cancer cell death induced by RT in recent years are primarily summarized in the current study. Among them, ferroptosis is an iron-dependent cell death strategy which is triggered by excessive lipid peroxidation. Notably, RT can induce ferroptosis by inducing the expression of acyl-CoA synthetase long-chain family member 4 (ACSL4) [9]. Simultaneously, the tumor protein p53 enhances RT-induced ferroptosis, partly by inhibiting the expression of solute carrier family 7 member 11 (SLC7A11) [10]. As a new form of immunogenic cell death program, necroptosis, which is regulated by Z-DNA-binding protein 1 (ZBP1)-mixed-lineage kinase domain-like pseudokinase (MLKL) signaling, has been found to improve the radiation-induced antitumor immunity of cells [11]. On the contrary, autophagy is more inclined to induce tumor resistance and is discussed in a subsequent section of this study. However, there is a need for further investigation to determine whether RT is able to induce pyroptosis or cuproptosis.

In general, there is a dose–effect relationship between the radiation dose and tumor control rate (that is, the increase in radiation dose can increase the killing effect of tumor cells within a certain scope). Most of our understanding of radiobiological principles is based on conventional photon radiotherapy (doses of 1.8–2.0 Gy per day) outcome studies. The most widely accepted model is the linear–quadratic (LQ) model. Currently, RT has entered the era of unconventional fractionated radiotherapy, including stereotactic body radiotherapy (SBRT), and FLASH RT. While the LQ model has been widely used for decades in radiation oncology, its applicability to hypofractionated RT remains controversial. Additional efforts are required to fully exploit the potential of these novel RT techniques to guide clinical interventions.

## 3. Molecular Mechanisms of Radioresistance in Tumor Cells

Radioresistance is a major impediment to therapeutic efficacy and results in tumor recurrence, as well as distant metastasis. Previous studies have recently reported on the mechanisms of radioresistance mediated by the tumor stroma [12,13]. The present study mainly reviews the radioresponse of resistant cancer cells (Figure 2). Moreover, the post-radiation-exposure tumor-induced immunosuppression effects are also summarized in a subsequent section of the current study.

### 3.1. Enhanced DNA-Repair Capability

RT kills tumor cells mainly by inducing double-strand DNA breaks. Therefore, an increased ability for DNA repair is strongly correlated with radioresistance in cancer cells. Serine proteinase inhibitor clade E member 2 (SERPINE2) promotes the repair potential of HR by activating the downstream repair protein RAD51 [14]. In a different study, the kinase ATM phosphorylates and stabilizes zinc finger E-box binding homeobox 1 (ZEB1). Then, ZEB1 binds to ubiquitin-specific peptidase 7 (USP7) and enhances its ability to deubiquitylate as well as stabilize the checkpoint kinase 1 [15]. In addition, pancreatic progenitor cell differentiation and proliferation factor (PPDPF), as well as Bromodomain-containing 4 (BRD4), can also enhance the repair capacity of tumor cells [16,17].

The deletion of leucine-rich repeat-containing protein 31 (LRRC31) can enhance DNA repair through genome-wide CRISPR library screening. This occurs by through regulation of the activity of DNA-PKcs and ATR-MSH2 signaling [18]. Previous studies have found that the repression of ubiquitin-conjugating enzyme E2O (UBE2O) or aurora kinase A (AURKA) can also promote radiation-induced DNA damage in lung cancer cells [19,20].

### 3.2. Cell-Cycle Checkpoint Activation

The integrity of DNA molecular structure and function is essential in the maintenance of the normal cellular function. When the DNA structure is damaged, the cell cycle of the affected tissue can be delayed or arrested to provide sufficient time for DNA repair. Therefore, tumor cells can enhance resistance to radiation by regulating cell-cycle arrest. For instance, cyclin K plays pivotal roles in the regulation of the responses that lead to DNA damage and control the G2/M checkpoint through modulation of the β-catenin/cyclin D1 axis [21]. A separate previous study has revealed that a positive feedback loop of the DNA-PK/AKT/GSK3β/cyclin D1 pathway can also activate the DNA-damage checkpoint [22]. Ubiquitin-conjugating enzyme E2T (UBE2T) interacts with and monoubiquitinates H2AX upon exposure to radiation, and hence, facilitates the activation of CHK1, as well as the cell-cycle arrest [23].

Topoisomerase IIβ-binding protein 1 (TopBP1) or Claspin is an important mediator in the checkpoint which increases the phosphorylation of CHK1 for the radioresistance of lung cancer cells [24]. In addition, RT can drive p21-activated kinase 1 (PAK1) to phosphorylate RAF1 on serine 338 and recruit checkpoint kinase 2 [25]. Notably, the HPV oncoproteins E6 and E7 promote the expression of the Ras-associated binding protein Rab12 to facilitate the arrest of G2/M [26]. Further, it has been shown that caspase-activated DNase (CAD) actively promotes self-inflicted DNA breaks, which leads to the arrest of the G2 phase in cancer cells, and hence, increases the survival of the cell after RT [27].

### 3.3. Cancer Stem Cells (CSCs)

Cancer stem cells (CSCs) represent a small population of cells (approximately 1~2%) in tumor tissue with characteristics of stem cells. Increasing evidence indicates that CSCs are closely related to the occurrence, treatment, prognosis, recurrence, and metastasis of tumors [28,29,30]. Fractionated irradiation promotes N-cadherin expression, which maintains the stemness of glioma stem cells by inhibiting Wnt/β-catenin signaling [31]. On the contrary, ribosomal S6 protein kinase 4 (RSK4) promotes CSC properties in esophageal squamous cell carcinoma (ESCC) by activating the Wnt/β-catenin pathway [32]. In colorectal cancer cells, JAK2/STAT3/CCND2 signaling contributes to cancer cell stemness and radioresistance [33]. In addition, it has been found that S100 calcium-binding protein A9 (S100A9) mediates the properties of CSCs in the brain microenvironment by activating the RAGE/NF-κB pathway [34]. It has been reported that with ultra-high doses of irradiation (~10^9^ Gy/s), the CSCs may enhance lysosome-mediated autophagy and decrease pyroptosis and apoptosis, as well as necrosis [35].

Pluripotency transcription factors, such as octamer-binding transcription factor 4 (OCT4), SRY-box transcription factor 2 (SOX2), and Nanog homeobox (NANOG), play an important role in the regulation of the self-renewal and differentiation of CSCs. It has been noted that THO complex 2/5 can facilitate the mRNA export and translation of SOX2 and NANOG [36]. Similarly, RAD51AP1 regulates the self-renewal of CSCs in breast cancer cells [37]. Additionally, the upregulation of ALG3 promotes the glycosylation of transforming growth factor beta receptor 2 (TGFBR2). This later activates the TGF-β/Smad pathway, which further promotes the expression of OCT4, NANOG, and SOX2 [38]. Moreover, the positive-feedback activation of ROS/AKT signaling contributes to the enrichment of CSCs in nasopharyngeal carcinoma [39].

### 3.4. Aberrant Metabolism

Deregulating cellular metabolism is a hallmark of cancer, and hence, can regulate the resistance of cancer cells to therapy [40]. For instance, the high expression of glutamine synthetase can promote nucleotide metabolism for the efficient repair of damaged DNA, and hence, contributes to the resistance of tumor cells against radiation [41]. In lung cancer, KEAP1/NFE2L2 mutations depend on glutamine metabolism to decrease intracellular ROS levels [42]. In addition, purine metabolism has been found to protect glioblastoma from radiation by promoting the repair of damaged DNA [43]. It has also been reported that metabolic pathways which are regulated by nicotinamide adenine dinucleotide (NAD^+^) and nicotinamide adenine dinucleotide phosphate (NADPH) can also potentiate radiation resistance in glioblastoma [44,45]. Moreover, oncostatin M receptor (OSMR) promotes oxidative phosphorylation by interacting with the NADH ubiquinone oxidoreductase core subunit S1/2 (NDUFS1/2) [46].

Recently, fatty acids have been found to be an alternative pivotal resource for energy, and thus, contribute to radiation-induced resistance by facilitating the oxidation of mitochondrial fatty acids (FAO) [47,48,49]. Notably, fatty acid metabolism can also enhance CD47-mediated anti-phagocytosis in glioblastoma multiforme. Mechanistically, acetyl-CoA, which is one of the main metabolites of FAO, upregulates the expression of CD47 by acetylating RelA K310 [47]. It is also notable that the crosstalk between lipid droplets and iron metabolism makes cancer cells more radioresistant [50]. In hepatocellular carcinoma (HCC) cells, the integration of glucose and cardiolipin anabolism, which is regulated by the mTORC1/HIF-1α/SREBP1 axis, promotes resistance to radiation by repressing cytochrome c extrusion [51].

### 3.5. Regulating the Activities of Autophagy

Autophagy represents a fundamental cellular metabolism that removes aberrant or redundant cellular contents. It also recycles metabolic substrates through lysosome-mediated degradation to maintain homeostasis processes [52]. Therefore, the autophagic response can protect cancer cells from RT-induced death [53]. For instance, it is evident that Annexin A6 (ANXA6) promotes autophagy by inhibiting the PI3K/AKT/mTOR axis in nasopharyngeal carcinoma tumor cells [54]. In HCC cells, it has also been found that early growth-response factor (Egr-1) facilitates irradiation-induced autophagy by transcriptionally activating Atg4B [55].

In addition, it is evident that mammalian STE20-like protein kinase 4 (MST4) increases autophagic activity by phosphorylating ATG4B in serine 383 in glioblastoma [56]. It has been shown that the depletion of SMAD family member 4 (SMAD4), which is a frequently mutated gene, can also induce high levels of autophagy in pancreatic cancer cells [57]. Further, autophagy inhibits type I IFN-dependent abscopal responses in breast cancer cells by limiting the accumulation of mitochondrial DNA (mtDNA) in the cytoplasm [58]. Generally, the inhibition of autophagy may promote the radiosensitivity of cancer cells, and hence, potentially improve the efficacy of RT.

### 3.6. Non-Coding RNA (ncRNA)

There are numerous unique ncRNAs that exist in cells. With the rapid development of sequencing technology, previous studies have enhanced the available understanding of ncRNAs. They are currently not termed as junk transcripts but are also functional molecules that can regulate cellular processes, including signal transduction, immune regulation, and therapy resistance [59]. Long non-coding RNAs (lncRNAs) are characterized as transcripts greater than 200 base pairs. LncRNA ANRIL promotes radiation resistance by regulating HR repair in lung cancer cells. Further studies indicate that lncRNA ANRIL interacts with the ATR protein to maintain its stability and protect it against ubiquitination-mediated degradation [60]. In addition, it is evident that LncAFAP1-AS1 can enhance the radioresistance of triple-negative breast cancer cells by regulating the Wnt/β-catenin pathway [61].

Evidently, lncRNAs are considered an important part of the competitive endogenous RNA (ceRNA) network. For instance, lncRNA HNF1A-AS1 binds with miR-92a-3p and enhances the radioresistance of non-small-cell lung cancer (NSCLC) by competitively regulating the MAP2K4/JNK axis [62]. Further, in radioresistant pancreatic cancer cells, the decreased expression of miR-23b increases autophagic activity via upregulation of the autophagy regulator ATG12 [63]. Additionally, exosomal miR-194-5p derived from irradiated dying tumor cells induces the arrest of the G1/S cell cycle and enhances the survival of tumor-repopulating cells by regulating the expression of E2F transcription factor 3 (E2F3) [64].

## 4. RT-Enhanced Tumor Metastasis

Tumor metastasis is the primary cause of failure in conventional cancer therapy [39]. Currently, the biological responses of normal tissues to RT-induced damage and the influence of them on metastatic proficiency are not fully understood. A recent study indicates that radiation exposure in healthy lung tissue induces a hospitable environment for metastatic growth. Mechanistically, RT induces neutrophil accumulation and activation in healthy lung tissue, leading to a range of tissue perturbations, including Notch activation [65]. Additionally, the epithelial–mesenchymal transition (EMT) is a biological process in which epithelial cells acquire mesenchymal characteristics, which makes them more invasive and metastatic. In cancer, this program is hijacked to endow cancer cells with tumor-initiating and metastatic potential [66,67]. For instance, RT can promote the expression of ADAM metallopeptidase domain 10 (ADAM10) in pancreatic cancer cells, leading to the cleavage of ephrinB2, which is expressed in stromal fibroblasts. Subsequently, the extracellular domain of ephrinB2 interacts with its receptor and drives EMT and invasion [68]. It has also been reported that RT can promote the stability of cell division cycle 6 (CDC6) protein, thereby promoting cell invasion and migration [69]. Moreover, the epithelial cell adhesion molecule (EpCAM) is associated with a hybrid epithelial–mesenchymal phenotype in breast cancer [70].

## 5. The Immune Effect of RT on Tumor

RT can also initiate anti-tumor immune responses. This is in addition to its direct killing potential of the tumor cells by damaging their DNA with high-energy rays. However, RT can also suppress anti-tumor immune responses in different ways to achieve tumor immune escape. Therefore, it is important to study and provide an understanding of the molecular regulatory mechanisms between RT and the host’s immune responses. This allows a combination of different treatments and improvement of the efficacy of RT.

### 5.1. RT Enhances Immune Responses

#### 5.1.1. Inducing the Type I Interferon Response

Type I interferon (IFN) is a pleiotropic cytokine with antiviral, antitumor, and immunomodulatory properties. Previous studies have shown that local RT can trigger the production of type I interferons, resulting in increased T-cell priming and the regression of the tumor [71]. Subsequent studies have also shown that the stimulator of interferon genes (STING) is essential for the induction of type I IFNs and the antitumor effect of RT. This mechanism appears to involve the cytosolic DNA sensor cyclic GMP-AMP synthase (cGAS) for sensing of DNA through DCs [72,73]. In parallel, RT can also lead to the accumulation of cytosolic micronuclei in cancer cells, which activates type I IFN through the cGAS/STING pathway and increases the infiltration of immune cells [74,75,76]. Recently, a study conducted by Deng et al. found that the ZBP1-MLKL necroptotic signaling pathway triggers type I IFN responses by inducing cytosolic mtDNA accumulation in irradiated cancer cells [11].

Retinoic acid-inducible gene I (RIG-I)-like receptors (RLRs) are key sensors of viral and host-derived RNAs, mediating the transcriptional induction of type I interferons, as well as other immune genes [77]. The family of RLRs encompasses RIG-I, melanoma differentiation-associated protein 5 (MDA5), and laboratory of genetics and physiology 2 (LGP2) [78]. Upon recognition of the radiation-induced endogenous double-stranded RNAs, RIG-I subsequently binds to the mitochondrial antiviral-signaling protein (MAVS), and eventually leads to the the activation of IFN-β [79]. Moreover, tumor-derived exosomes produced following RT can also transfer immunostimulatory RNA to DCs and stimulate the cGAS-STING-dependent activation of type I IFNs [80].

#### 5.1.2. Increasing Tumor Antigen Presentation

Previous studies have demonstrated that CD8^+^ T cells are an important part of tumor immunity and CD8^+^ T-cell infiltration is correlated with better survival in patients with cancer. Radiation can increase major histocompatibility complex I (MHC-I) expression on the surface of tumor cells and expand the intracellular peptide repertoire, which are represented by MHC-I molecules [81,82]. It has been reported that ablative radiation with between 15 and 20 Gy dramatically promotes T-cell priming, leading to local or distant tumor regression in a CD8^+^ T cell-dependent manner [83]. Further, neoantigens that arise as a result of tumor-specific mutations are known to be highly immunogenic and represent ideal targets for T cells [84]. Exposure to radiation can also stimulate the production of subclonal neoantigens that can contribute to the synergy between RT and immunotherapy [85,86]. For instance, functional analysis in a patient with NSCLC revealed that there is rapid in vivo expansion of CD8^+^ T cells that recognize a neoantigen which is encoded by karyopherin α2 (KPNA2) [87,88]. Elsewhere, it has been found that radiation-induced neoantigens may be represented by MHC-II to enhance the activation of CD8^+^ T cells [89].

#### 5.1.3. Release of DAMPs from Tumor Cells

Recent studies show that local RT creates an inflammatory microenvironment which is characterized by release of tumor antigens and DAMPs, which include high-mobility group protein B1 (HMGB1), adenosine triphosphate (ATP), and calreticulin [90]. Calreticulin acts as an “eat me” signal to mobilize antigen-presenting cells (APCs) [91]. Further, HMGB1 released from the tumor cells promotes antigen cross-presentation of DCs via the activation of TLR4-MyD88 [92]. On the other hand, the release of ATP from the dying cancer cells activates P2X7 purinergic receptors on DCs and triggers the secretion of interleukin-1β(IL-1) through the activation of caspase-1, which depends on NLR family pyrin domain containing 3 (NLRP3) [93]. These findings show that the therapeutic efficacy of RT may partly depend on DCs to prime tumor-specific CD8^+^ T lymphocytes.

#### 5.1.4. Enhancing T-Cell Infiltration

Parallel to the effects on DC activation, the increased secretion of cytokines also contributes to the infiltration of cytotoxic T cells in response to RT [94]. Ionizing radiation markedly increases the production of CXC chemokine ligand 16 (CXCL16) and induces strong chemotaxis of tumor-specific CD8^+^ T cells [95]. Furthermore, the expression of CXCL9, CXCL10, and CXCL11 is upregulated by RT and is involved in the chemoattraction of activated CD8^+^ T cells into the tumor microenvironment [96]. Otherwise, natural-killer group 2 member D (NKG2D) is an important activating immune cell receptor and is expressed on natural-killer (NK) and T cells. Irradiation could increase the expression of NKG2D ligands (NKG2DL) in tumor cells, which promotes immune cell-mediated cytolysis without prior sensitization and MHC restriction [97,98].

Growing evidence suggests that the type of dose fractionation regimen has an effect on immune infiltration. Compared with the two fractionation regimens of 20 Gy×1 and 6 Gy×5, three fractions of 8 Gy are more effective in the induction of the “abscopal effect”, referring to tumor regression observed outside of the radiation field, and tumor-specific T cells in combination with CTLA-4 blockade [99]. Low-dose radiation remodels the tumor microenvironment and predominantly triggers CD4^+^ T-cell infiltration together with immune-checkpoint blockade [100,101]. High-dose RT with a single radiation dose of 30 Gy could result in potent CD8^+^ T-cell infiltration and a decrease in myeloid-derived suppressor cells (MDSCs) [102]. However, high-dose RT can also cause harmful side effects, such as lymphodepletion at primary sites, increased local and circulating MDSCs, and upregulated regulatory T cells (Tregs). Therefore, Welsh et al. proposed a new radiation strategy in which high-dose RT is used to prime T cells and low-dose RT delivered to secondary tumors is applied to modulate the stroma and improve the infiltration of immune cells into secondary tumors [103].

### 5.2. RT-Induced Tumor Immune Escape

#### 5.2.1. Negative Regulation of the Type I Interferon Response

As discussed earlier, the type I IFN pathway is essential for the activation of DC, T-cell priming, and tumor regression. However, the DNA exonuclease Trex1, induced by radiation doses above 12 Gy, is sufficient to degrade the cytosolic DNA of the irradiated tumor cells, and hence, attenuate the production of IFN-β [104]. In addition, it is evident that Caspase-3 inhibits the accumulation of double-stranded DNA in the cytosol of the irradiated cells, and this results in impaired secretion of type I IFN as well as abscopal responses to RT [105]. Further, it has been shown that tumor cells can also hijack caspase-9 signaling to weaken radiation-induced immunity through the mtDNA-cGAS-STING pathway [106]. Similarly, it has been found that the ablation of caspase-8 promotes the ZBP1-MLKL necroptotic pathway and increases the enrichment of cytosolic mtDNA [10,107].

In addition, laboratory of genetics and physiology 2 (LGP2) can also negatively regulate the RNA-sensing MDA5 and RIG-I pathways by interfering with TRAF ubiquitin ligase [78,108]. Ataxia telangiectasia mutated (ATM) is an apical kinase which is responsible for the radiation-induced DNA damage response and restrains tumoral immunogenicity using interferon signaling [109]. Analogously, ATR inhibition can significantly potentiate the interferon response by activating cGAS/STING signaling and the RNA sensing pathway, which is MAVS-dependent [110]. However, there is a notable finding on the negative regulation of type I IFN expression by the non-canonical NF-κB pathway. Mechanistically, irradiated tumor cells also inhibit the binding of transcription factor RelA to the IFN-β promoter in DCs [111].

#### 5.2.2. Overexpression of Immunosuppressive Molecules

Although immunotherapy can improve the efficacy of RT, tumor cells can also produce immunosuppressive factors and achieve immune escape. Transforming growth factor beta (TGFβ) acts as a multipotent cytokine and suppresses the expression of chemokine receptor CXC motif 3 (CXCR3) in CD8^+^ T cells. This is through the promotion of the binding potential of Smad2 to the CXCR3 promoter [112]. The high expression rate of the cluster of CD47 in tumor cells can also bind to signal regulatory protein α (SIRPα) on macrophages, and hence, cause inhibition of the phagocytic capacity [113]. Furthermore, the cancer cells continuously export cGAMP to the extracellular space when treated with RT, which results in the constitutive activation of the cGAS-STING pathway. However, ectonucleotide pyrophosphatase phosphodiesterase 1 (ENPP1) can also selectively degrade extracellular cGAMP and generate extracellular adenosine [114,115].

RT also increases the expression of the adenosine-generating enzyme CD73 in irradiated breast cancer cells. This can, in turn, limit the infiltration of conventional dendritic cell type 1 and T cells in to the tumors [116]. Remarkably, the cancer cell-intrinsic cGAS-STING pathway, which is activated by RT, promotes immune evasion by upregulating programmed cell death-ligand 1 (PD-L1) [117]. In a recent study, it has been found that the lactate derived from tumor cells activates MDSCs through the GPR81/mTOR/HIF-1a/STAT3 pathway and is necessary for the radioresistance of pancreatic cancer [118].

#### 5.2.3. Recruitment of Immunosuppressive Cells

After radiation, different inhibitory immune cells are recruited to the tumor microenvironment. Therefore, the cells counterbalance the therapeutic effect of RT at the tumor region. Among the recruited inhibitory immune cells are tumor-infiltrating myeloid cells and Tregs, which play crucial roles in remodeling the immunosuppressive tumor microenvironment. For instance, C-C motif chemokine ligand 2 (CCL2) is upregulated in tumor cells following RT, and hence, leads to the corecruitment of tumor-associated macrophages, as well as Tregs, through the CCL2/chemokine receptor 2 (CCR2) axis [119,120]. Similarly, C-C motif chemokine ligand 2 (CCL5) can also promote the infiltration of immunosuppressive CCR5^+^ monocytes into the tumor site [121]. There is a need for further research on how radiation promotes the expression of CCL2 and CCL5. In prostate cancer, colony-stimulating factor 1 (CSF1) significantly enhances the recruitment of tumor-associated macrophages (TAMs) and MDSCs. Previous mechanistic investigations have shown that ABL proto-oncogene 1 (ABL1) binds to the promoter region of CSF1 and upregulates the CSF1 gene after irradiation [122].

Collectively, a combination of immunotherapy and RT has risen to the forefront of cancer research studies and is being rapidly integrated into oncology. When combined with immunotherapy, the primary consideration for irradiation dose is to maximize synergy with immunotherapy rather than maximize DNA damage on the tumor cells. However, more studies are needed to explore better immunotherapy regimens for the different types and stages of cancer. In addition, elaborate exploration of the dose, fractionation, and sequence of RT, as well as better control of the side effects of the combination therapy, will allow patients to benefit more from the synergy of RT with immunotherapy.

## 6. Conclusions and Perspectives

More than a century after its discovery, RT has effectively achieved the local control of tumors but has a limited effect on the overall survival of patients. This has been a great challenge in the field of RT. Clinically, intrinsic or acquired resistance to RT is a primary obstacle to the desired suppression of tumors. There are several pathways involved in the resistance of cancer cells to radiation. The pathways include DNA repair, cell-cycle checkpoints, and metabolic pathways. Therefore, there is a need to develop novel radiosensitizing drugs and new combination therapies for cancer.

Radiation oncology is facing a new era of development with the emergence of new RT technologies such as SBRT, heavy-particle-beam therapy, proton-beam RT, and FLASH RT. However, there are still numerous unresolved issues regarding the radiobiological effects and therapeutic schedule of the new techniques, which require the collaboration of radiobiologists, radiophysicists, and radiologist to obtain more convincing solutions. Furthermore, tumors often shift during RT because of the movement, filling, or emptying of the organs in which they are located. Consequently, more precise image-guided techniques are needed to maximize the killing effect and minimize off-tumor cytotoxicity. Additionally, internal radionuclide therapy is becoming a promising therapy for the treatment of tumors. However, the development and application of radiopharmaceuticals are considerably impeded by the limited supply of medical radionuclides. In the future, technological advances in accelerators and nuclear reactors will be needed to solve the supply problem. Further, more targeted radiopharmaceuticals should be developed.

## Figures and Tables

**Figure 1 biomolecules-12-01167-f001:**
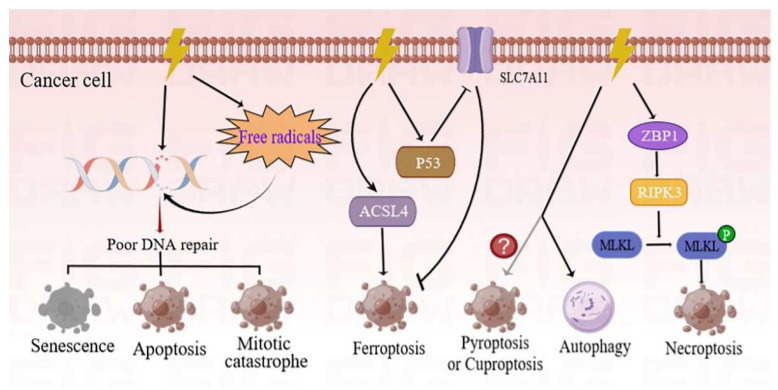
Different forms of cancer cell death programs induced by RT. RT can directly damage the DNA through production and deposition of ionizing energy or indirectly via free radicals. The DNA strands can be damaged in single-strand and double-strand breaks. If these DNA lesions cannot be properly repaired, tumor cells will initiate different types of death programs, including mitotic catastrophe, apoptosis, and senescence. Ionizing radiation (IR) can induce not only the expression of ACSL4 but also the activation of p53, and hence, results in elevated ferroptosis. The ZBP1-RIPK3-MLKL necroptotic cascade induces accumulation of cytoplasmic mtDNA in irradiated tumor cells, and consequently, activates an anti-tumor immune response. Conversely, autophagy is more inclined to cause cancer cell survival following IR treatment. However, there is a need for further studies to determine whether RT can induce pyroptosis or cuproptosis. ACSL4—acyl-CoA synthetase long-chain family member 4; p53—tumor protein p53; ZBP1—Z-DNA-binding protein 1; RIPK3—receptor-interacting serine/threonine kinase 3; MLKL—mixed-lineage kinase domain-like pseudokinase; mtDNA—mitochondrial DNA.

**Figure 2 biomolecules-12-01167-f002:**
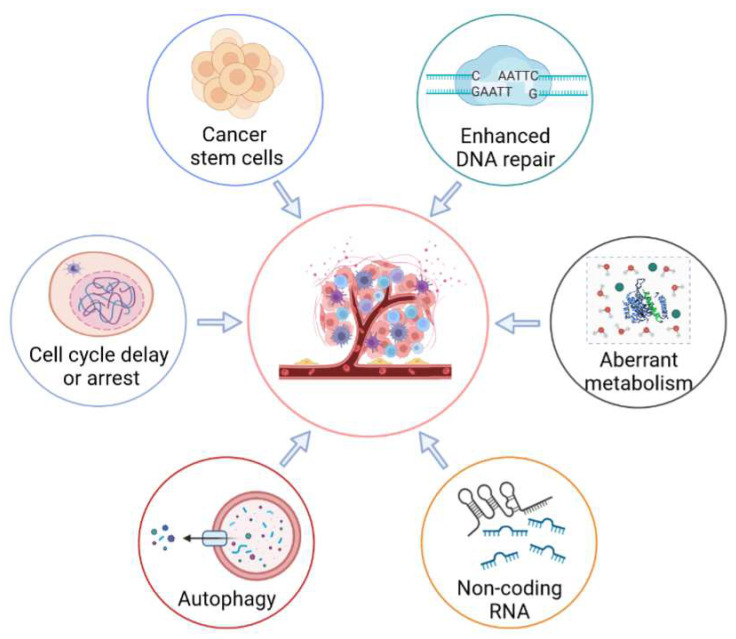
Molecular mechanisms of radioresistance mediated by tumor cells. Radioresistance of tumor cells is a major barrier to successful cancer treatment. In order to evade cell death, tumor cells have developed a variety of strategies, including enhancing DNA repair, activating cell-cycle checkpoints, regulating the self-renewal and differentiation of cancer stem cells, enhancing cellular metabolism, regulating the activities of autophagy, and regulating the expression of non-coding RNA.

## Data Availability

Not applicable.

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
