# Peer review of "Deciphering the Biological Effects of Radiotherapy in Cancer Cells"

_biomolecules, 2022, doi:10.3390/biom12091167_

Round 1
Reviewer 1 Report
This manuscript is a literature review about the effects of radiation on cells in the context of radiotherapy. The title is very good and reflects the intention of the authors to explore the mechanisms of radiation-induced biological changes in cells, which is of course an important medical issue. However, the content focuses mainly on the aspects of resistance to radiotherapy, which have been covered in other review articles. I would suggest the authors to revise the manuscript as follow, which may make this article to serve better for the need in the research community.
1. Add more information or as a separate paragraph on the topic that radiation may induce the cells to acquire properties for metastasis.
2. Discuss briefly about the types of radiation therapy including systemic radiotherapy.
3. This review is mainly from the external radiotherapy point of view. Is there anything different regarding systemic radiotherapy-induced effects on cells?
4. In the conclusion part, the authors mentioned about “tumor shifting”, it is good to describe a bit more on which kind of shifting the tumor has or the authors meant to say.
Author Response
Dear reviewer,
We feel great thanks for your professional review work on our article. As you are concerned, there are several problems that need to be addressed. According to your nice suggestions, we have made extensive corrections to our previous manuscript. The detailed corrections are listed below.
Point 1: Add more information or as a separate paragraph on the topic that radiation may induce the cells to acquire properties for metastasis.
Response 1: Thanks for your insightful suggestions. Indeed, radiotherapy can induce tumor cells to acquire properties for metastasis. We added a separate paragraph in the revised manuscript. Please see page 6 of the revised manuscript, line 222–239.
Point 2: Discuss briefly about the types of radiation therapy including systemic radiotherapy.
Response 2: Thanks for your kind advice. Radiation therapy can be divided into external beam radiotherapy and internal irradiation according to the radiation modality. Currently, internal radiotherapy is mainly focused on radionuclide therapy. Meanwhile, most of the radionuclide therapy is a form of systemic radiotherapy. Our article has described the developments in external irradiation therapy in recent years. Therefore, We briefly discuss the problems and prospects of radionuclide therapy in the last part of the revised manuscript.
Point 3: This review is mainly from the external radiotherapy point of view. Is there anything different regarding systemic radiotherapy-induced effects on cells?
Response 3: This is an excellent question. In systemic radiotherapy, more cells are irradiated than in local radiotherapy. In our perspective, these cells may have an effect on each other, which in turn causes complex biological effects. Regarding the question you raised, subsequent experiments are needed to obtain convincing conclusions.
Point 4: In the conclusion part, the authors mentioned about “tumor shifting”, it is good to describe a bit more on which kind of shifting the tumor has or the authors meant to say.
Response 4: Thanks for your comments. Since we did not express it clearly, we are sorry for your misunderstanding. We have revised in conclusion part of the manuscript as: “ Furthermore, tumors often shift during radiotherapy RT because of the movement, filling, or emptying of the organs in which they are located. Consequently, more precise image-guided techniques are needed to maximize the killing effect and minimize off-tumor cytotoxicity.”.
Thank you again for your positive comments and valuable suggestions to improve the quality of our manuscript.
Yours sincerely,
Zhou Lu
Reviewer 2 Report
The authors reviewed the topics comprehensively.
Major comment.
Page 7, line 297-304. Of course, you mentioned immunosuppressive effect by high does RT later, it would better to describe it a little here.
Minor comments.
1. In the first page you used radiation therapy(RT). From the second page you used radiotherapy. I recommned to use RT or radiotherapy, not mix.
2. Page 2, line 48. Previous -->previous
3. Fig 1. mitotic catastrope --> Mitotic catastrope
4. Fig 2. cell cycle delay --> Cell cycle delay
Author Response
Dear reviewer,
We very much appreciate the time and effort you’ve put into your comments. We really appreciate the your insightful suggestion. We have carefully reviewed the comments and have revised the manuscript accordingly. The detailed point-by-point responses are listed below.
Point 1: Page 7, line 297-304. Of course, you mentioned immunosuppressive effect by high does RT later, it would better to describe it a little here.
Response 1: We appreciate the reviewer’s insightful suggestion and agree that it would be useful to improve the quality of our article. Therefore, we added the following sentence to page 8, line 319-321 in the revised manuscript: “However, high-does RT can also cause harmful side effects, such as lymphodepletion at primary sites, increased local and circulating MDSCs, and upregulated regulatory T cells (Tregs).”.
Point 2: In the first page you used radiation therapy(RT). From the second page you used radiotherapy. I recommned to use RT or radiotherapy, not mix.
Response 2: We sincerely thank the reviewer for careful reading. As suggested by the reviewer, we have corrected the “radiotherapy” into “RT” from the second page.
Point 3: Page 2, line 48. Previous -->previous.
Response 3: We were really sorry for our careless mistakes. Thank you for your reminder.
Point 4: Fig 1. mitotic catastrope --> Mitotic catastrope.
Response 4: We feel sorry for our carelessness. In our resubmitted manuscript, the typo is revised. Thanks for your correction.
Point 5: Fig 2. cell cycle delay --> Cell cycle delay.
Response 5: Thanks for your careful checks. We have carefully checked the manuscript and corrected the errors accordingly.
Once again, thank you very much for your comments and suggestions.
Yours sincerely,
Zhou Lu